# Novel Perspective Coatings for the Optoelectronic Elements: Features of the Carbon Nanotubes to Modify the Surface Relief of BaF$_2$ Materials

**Natalia Kamanina** [1,2,*], **Pavel Kuzhakov** [1] **and Dmitry Kvashnin** [3,4,5] 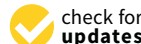

[1] Lab for Photophysics of Media with Nanoobjects at Vavilov State Optical Institute, Kadetskaya Liniya V.O., dom 5/2, 199053 St.-Petersburg, Russia; kpv_2002@mail.ru

[2] Electronics Department, St.-Petersburg Electrotechnical University ("LETI"), Ul.Prof.Popova, dom 5, 197376 St.-Petersburg, Russia

[3] Emanuel Institute of Biochemical Physics RAS, 4 Kosigina st., 119334 Moscow, Russia; dgkvashnin@phystech.edu

[4] Electronics Department, National University of Science and Technology "MISiS", Leninskiy Prospect 4, 119049 Moscow, Russia

[5] School of Chemistry and Technology of Polymer Materials, Plekhanov Russian University of Economics, Stremyanny Lane, 36, 117997 Moscow, Russia

* Correspondence: nvkamanina@mail.ru; Tel.: +7-(812)-327-00-95

**Abstract:** It is well known that the optimization of the basic properties of materials is related not only to changes of the substance of the material itself, but can also predict the change of their surface. In this regards, the search for, and study of, new nanostructured coatings based on the laser deposition method becomes extremely promising. Here, we used a laser-oriented deposition technique in order to place carbon nanotubes in a vertical position on the BaF$_2$ surface to modify it. Such modification affected the increasing material transparency, connected with a decrease of the reflection via change the Fresnel losses; hydrophobicity and microhardness as well. Characteristics of the obtained material were studied via spectral analysis, AFM-method, wetting-angle measurements, microhardness estimations to support the possible covalent bonding between the carbon atoms and the interface materials atoms. Moreover, the quantum–chemical calculations completely confirmed the experimental results of the changes of electronic properties of BaF$_2$ substrate after deposition of the CNTs. As the results novel optimized structure based on BaF$_2$ is presented to be used in general optoelectronics, cosmos and laser technique as well.

**Keywords:** inorganic materials; barium fluoride crystal; carbon nanotubes; interface; laser–matter interaction

## 1. Introduction

Over the last two decades, the improvement of the properties of the optoelectronic materials—as well as the influence of the effective nanoparticles—on the spectral, mechanical and wetting features is being a studied with extreme intensity. These parameters are associated with the expansion of the absorption spectral range, changes of the system electronic structure, and are also responsible for the manifestation of a tendency to increase the hydrophobicity and the durability of optoelectronic materials. Among these novel nanoparticles the carbon nanotubes (CNTs), reduced graphene oxides and nanofibers occupy the special place [1–9]. The basic features of these nanostructures are related to their high conductivity, strong hardness of their C–C bonds, as well as the complicated and unique mechanisms of charge carrier transfer. These nanoobjects can dramatically vary not only

the surface/interface of the inorganic/organic systems but can modify the basic physical and chemical characteristics of the whole materials as well.

It is worth mentioning that choosing these nanoobjects to improve the properties of inorganic materials; it is worth paying specific attention to their mechanical, refractive and hydrophobic properties. It should be remarked that Young's modulus of the carbon nanotubes is close to 0.32–1.47 TPa, which has been verified in the papers [10,11]. High conductivity of graphene oxide, as excellent electrode materials and its features as the unique 2D crystals for the ultrafast photonics have been shown previously [12,13]. Increase of the refractivity of the organics doped with nanofibers has been presented as well [14]. Thus, different extended area of the application of the inorganic/organic systems can be considered with good advantage using the carbon nanotubes, graphene and nanofibers as the excellent modifiers.

Here the solid substrate from $BaF_2$ inorganic crystal operated in the UV-Vis-IR spectral range is considered as the good model inorganic matrix to predict the change of the basic properties via the CNTs laser-oriented deposition (LOD) method.

Let us to pay attention to the inorganic crystal of barium fluoride. These materials have been studied by different scientific teams. Thus, the dependence of the intensity of the emission of self-trapped excitons, cross-luminescence and intra-band luminescence on the excitation density in $BaF_2$ by electron pulses with the pulse width of three nanoseconds at 300 keV has been studied in the article [15]. These results allowed, among other things, to develop the concept of creating and developing a new detector based on the barium fluoride materials [16]. Moreover, the $BaF_2$ materials can be used as the dopant. Influence of fluoride modifier on luminescence properties of the rare earths in different glass hosts has been examined in the paper [17]. The excitation and emission spectra of $Pr^{3+}$ and $Er^{3+}$ ions in the studied glasses have been registered under the $BaF_2$ introductions. However, previous studies have not addressed the investigation of the strength, refractive and hydrophobic properties of barium fluoride crystal when modifying its surface by the CNTs placement.

Naturally, CNTs can be applied to the surface of materials by different methods. CVD, PVD, pyrolysis and other approaches are used. Each approach has its own advantages and disadvantages [18,19]. However, the choice of the LOD technique is due to the fact that it differs significantly from the traditionally used CVD and PVD coating deposition methods. When using LOD technology, there is no great loss of substance and its application is oriented, that is, in a certain direction. In addition, the substrate temperature can be no more than 80 degrees, which also differs in a positive way from the known methods mentioned above.

Different experimental investigations and quantum–chemical calculations are coincided to each other in order to support the considered idea and reveal the change of the spectral, wetting angle and microhardness of the $BaF_2$ materials via nanotechnology approach.

## 2. Materials and Methods

Due to the transparency for the light from the ultraviolet radiation to infrared one, lenses and prisms for the IR optics and lasers are made from $BaF_2$ single crystals. Moreover, because of its anisotropy of the third-order nonlinear optical susceptibility, the $BaF_2$ crystal is widely used to improve the time contrast and other characteristics of the femtosecond laser pulses during the nonlinear process. Furthermore, $BaF_2$ can be used as the output window to protect the optoelectronic devises from the dust in the atmosphere and from the high density irradiation in the optoelectronics photodetectors used in the complicated technical schemes. Thus, the study and modification of the properties of this crystal is very important. It should be noticed that for the current experiments the $BaF_2$ substrate with the crystal lattice parameter close to 6.2 Å was purchased from the TydexOptics Co. Ltd. (St. Petersburg, Russia). The thickness of the $BaF_2$ substrate was 5 mm and the diameter was 35 mm. A sample of barium fluoride in the task bar and the type of the CNTs used is shown in Figure 1.

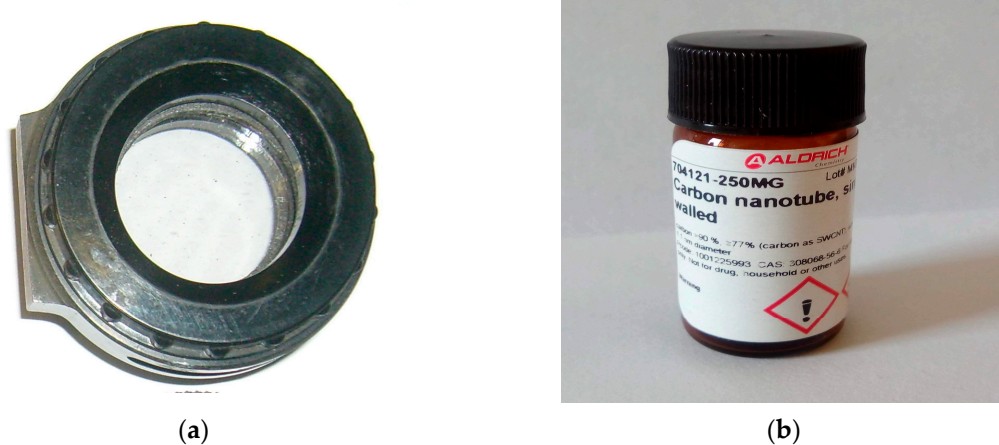

(**a**)                    (**b**)

**Figure 1.** (**a**) Sample of the BaF$_2$ material treated and (**b**) type of the carbon nanotubes (CNTs).

To modify the considered material's surface the IR CO$_2$ laser operated at the wavelength of 10.6 µm with the power of 30-W and a beam diameter of 5 mm was used. This laser was connected with the vacuum chamber. The CNTs were placed at the materials interface under the conditions when an additional electric field of 100–600 V·cm$^{-1}$ was applied in order to orient the nanotubes in the vertical position during the deposition process. Thus, the LOD method was realized efficiently. The general view of the applied block scheme is shown in study [20]. Some view of the block scheme of the experimental deposition technique is shown in Figure 2.

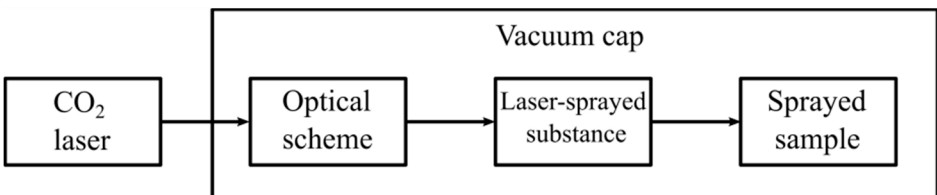

**Figure 2.** Principle scheme of laser–deposition technique.

One can see that the laser system is connected with a vacuum hood, which contains the fixing unit samples and the additional electric setup for placing substances deposited orthogonally on the substrate. It should be repeated once again that to modify the properties of the inorganic materials via their surface treatment using the LOD method [21,22], the single wall carbon nanotubes (SWCNTs) type #704121—with the diameter placed in the range of 0.7–1.1 nm—purchased from Aldrich Co. Ltd. (Karlsruhe, Germany) were used. It should be remarked that the dimension of the carbon nanotubes is important in order to combine the CNTs diameter directly with the elementary lattice parameter of the chosen model matrix material. Thus, it is very important to orient the CNTs via electric field applying, as is tentative and fundamental shown in Figure 3a. The AFM image of the sample treated is shown in Figure 3b as well.

The spectra of the nanoobject-treated materials were measured using the IR Furrier FSM-1202 instruments ("Modern Lab.", Moscow-St-Petersburg-Ekaterinburg, Russia) as well as using the VIS SP-26 spectrophotometer operated in the spectral range of 250–1200 nm. POLAM-P312 microscope ("LOMO", Saint-Petersburg, Russia) was applied to make the image of the materials treated. The microhardness was measured via using the PMT-3M device produced by "LOMO" plant (St. Petersburg, Russia) with the ability to vary an indenter force as well. The special accent was given to observe the relief at the material surface via checking the wetting angle. In this case, the camera with parameters as compact F1.6 1/3 CS mount 6.0–60 mm manual focal iris zoom lens for CCTV camera (black) was applied. Moreover, the OCA 15EC device was purchased from LabTech Co. Ltd. (St. Petersburg–Moscow, Russia) and was used to carefully control the wetting-angle change too.

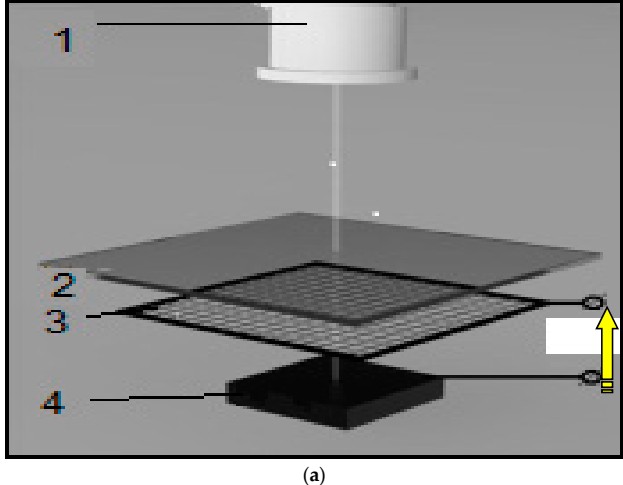

(a)

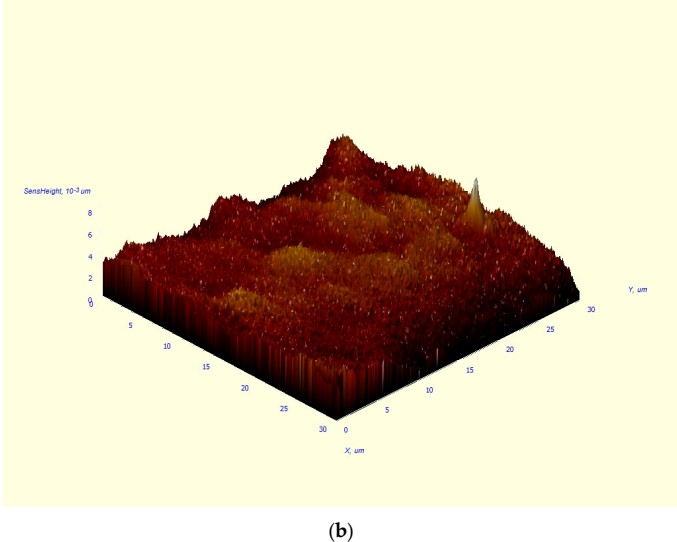

(b)

**Figure 3.** Possible evidence to orient the CNTs. (**a**) 1—$CO_2$ laser, 2–substrate (samples), 3—metal grid, 4—carbon target; (**b**) view of the treated sample obtained via AFM. Axes in microns (*x* and *y*) and microns $\times 10^{-3}$ at *z* axes.

The make the modified surface analysis, an atomic force microscope AFM Solver Next (purchased from NT MDT Co. Ltd., Zelenograd, Moscow Region, Russia) was applied. It should be mentioned that it permits to register the homogeneity of the surfaces and to estimate their roughness under the condition to use the nanostructuring process as well. It should be noticed that, of course, the AFM does not unconditionally confirm the penetration of CNT into the near-surface layers of the sample. A more detailed study is required, for example, of radiographs, etc., which will be carried out in subsequent experiments not only with barium fluoride, but also with other crystals functioning in the UV-IR spectral regions.

## 3. Results and Discussion

The change of the transmittance spectra in the range of 250–450 nm and in the range of 2–10 microns of the $BaF_2$ + CNT are shown in Figure 4. It was observed the strong transparency increasing in the range of the wavelength of 280–380 nm and of 2–3% transparency change in the range of 2–10 microns. It should be remarked that the increase of the transparency could be explained by the decrease of the Fresnel losses. Really, it may be because of the refractive index of the CNTs was close to 1.1 that was substantially less than for the nanostructured $BaF_2$ materials (*n*~1.38–1.39 at the operated wavelength of 10.6 μm). Thus, the analytical calculation via the fundamental Fresnel formulas [23] could predicts

the change of the reflection up to 1–2%, which coincided with the increased experimental value of the transparency up to 2% and not less after the surface structuration of the $BaF_2$ substrate. Data presented in Figure 4c is regarded to the reflection parameters change. One can see that these data coincided with the theoretical Fresnel losses estimated by the analytical calculations.

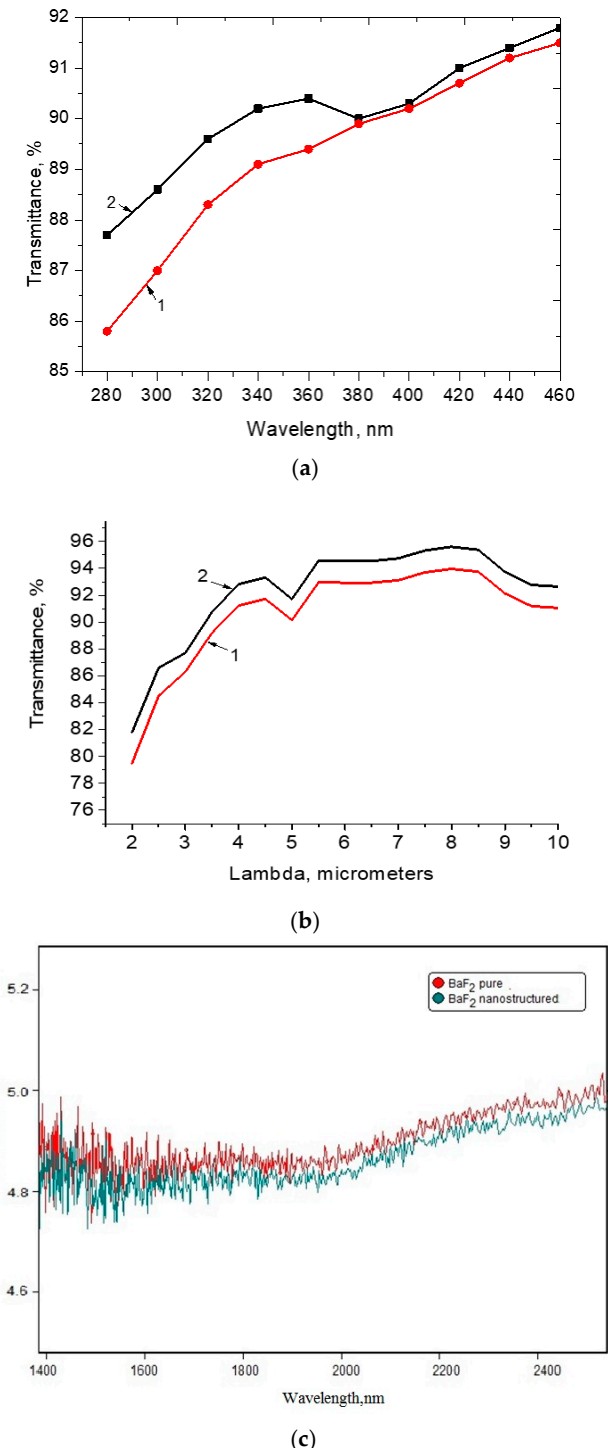

**Figure 4.** (**a**) UV-Vis transmittance spectra for the pure and structured $BaF_2$ crystals; (**b**) IR transmittance spectra for the pure and structured $BaF_2$ crystal; Curve 1 indicates the pure materials; curve 2 regard to the $BaF_2$ + CNTs materials; (**c**) reflection spectra for the pure (upper red curve) and modified $BaF_2$ (the lower blue curve) for the near IR range.

Really, let us show how the coating layer is formed, taking into account the minimization of Fresnel losses, in the classical case. For example, as the good evidences, that the Fresnel losses can be changed, which can be easy shown based on the classical glass substrate, which is basically used as the etalon. Really, using the classical method in order to reduce the losses at the reflection process and thus to improve the light transmission (aperture ratio) of the optics, the surface of the glass can be subjected to special treatment, which is called as the "enlightenment of optics". On the surface of the matrix materials (glass) the thin film should be applied, the refractive index of which should be less than the refractive index of glass, namely: $n_{\text{film}} = \sqrt{n_{\text{glass}}}$. In order to make the minimal reflection losses, the film must have a certain thickness, which can be calculated by the formula: $h = \frac{d}{n_{\text{film}}}$, there $h$—the geometric thickness of the film, $d$—the optical thickness of the film obtained as $d = \frac{\lambda}{4}$, $\lambda$—the wavelength of the light in that part of the spectrum, where it is necessary to obtain the maximum of the transmittance. It is well known that the film with the thickness of $\frac{\lambda}{4}$ from the substance with the refractive index of $\sqrt{n_{\text{glass}}}$ decreases the reflective coefficient dramatically. If one considers the interface such as the air–glass with the reflective index of the glass material closed to 1.5, the reflection from one surface of the substrate is approximately 4% and from two surfaces it is approximately 8%.

Let us return to the consideration of barium fluoride materials. It should be mentioned that using the innovative structuration approach, namely proposed LOD technique and possible (as a forecast stage) forming the covalent bonding between the CNTs (with the little refractive index $n$ of ~1.1) and near-surface atoms of $BaF_2$ materials (with the refractive index of $n$~1.38–1.39), the Fresnel losses via the reflection can be changed, at least on some percent. For example, for use in the IR range of this material, even changing the reflection loss in the region of one percent is a good achievement. In addition, if the usual classical coating procedure requires applying from 10 to 30 layers, then using the technology shown in this work, it is enough to conduct one treatment cycle.

Moreover, it is important to mention that the change of the spectral characteristics is coincided with the change of the microhardness of the $BaF_2$ material. The mechanical parameters change is shown in Table 1. It should be noted that the microhardness is increased on 10% and more for the $BaF_2$ materials, which surfaces were treated with the CNTs via LOD procedure. The 2 g indenter was used. Indeed, the dramatic macrohardness increase is possible due to very large Young's modulus of the CNTs and their ability to be incorporated in the material surface. It should be noted that the strength of the coatings remained at the same level when testing previously treated materials were made after 5 years.

**Table 1.** Measured microhardness parameters.

| Materials | Value of the Microhardness, Pa $\times 10^9$ | | | | |
|---|---|---|---|---|---|
| Pure $BaF_2$ | 0.0762 | 0.0762 | 0.0762 | 0.0797 | 0.0797 |
| $BaF_2$ + CNTs | 0.0918 | 0.0965 | 0.0875 | 0.0835 | 0.1015 |

One of the most important influences of CTN on the $BaF_2$ is the changes of the wetting angle after CNT deposition (Figure 5). It is clearly observed that deposition of the CNTs on the $BaF_2$ surface leads to increasing of the wetting angle from 56° up to 67°. It should be mentioned that we have tested 10 samples of the $BaF_2$ substrate. The error of the experiment was about 5%. The error is estimated as a deviation of each measured parameter from the average.

Analyzing the obtained results we have considered the idea to form the covalent bonding between the incorporated CNTs with the promising diameter and the $BaF_2$ atoms from the elementary crystal cells. This idea was slightly supported by the quantum–chemical calculations.

Using the density functional theory, the influence of the carbon nanotubes deposited on the $BaF_2$ surface to its electronic properties was estimated. In Figure 6 the atomic structure of the studied interface is shown. We choose the (100)-$BaF_2$ surface, because this surface termination contains two types of the atoms (Ba, F) on the surface. Due to the big amount of the atoms in the considered unit cell of the substrate, the CNT has quite short length of 1.2 nm. Upper end of the tube was passivated by

the hydrogen atoms. After the deposition of the CNT on the BaF$_2$ surface the main changes appear in the region close to the BaF$_2$/CNT interface. Thus, the chosen length of the nanotube is enough for the qualitative description of the changes in the electronic properties. Studied system consists of the 105 barium atoms, 210 fluorine atoms and 110 carbon atoms, which is the CNT together with the 10 hydrogen atoms (passivation of the upper end).

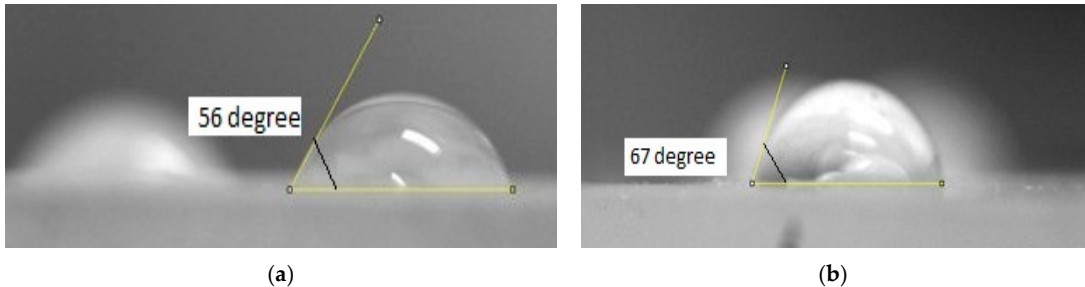

(**a**) (**b**)

**Figure 5.** Comparative wetting-angle measurements. (**a**) Wetting angle for the pure BaF$_2$ substrate; (**b**) wetting angle for the structured BaF$_2$ substrate.

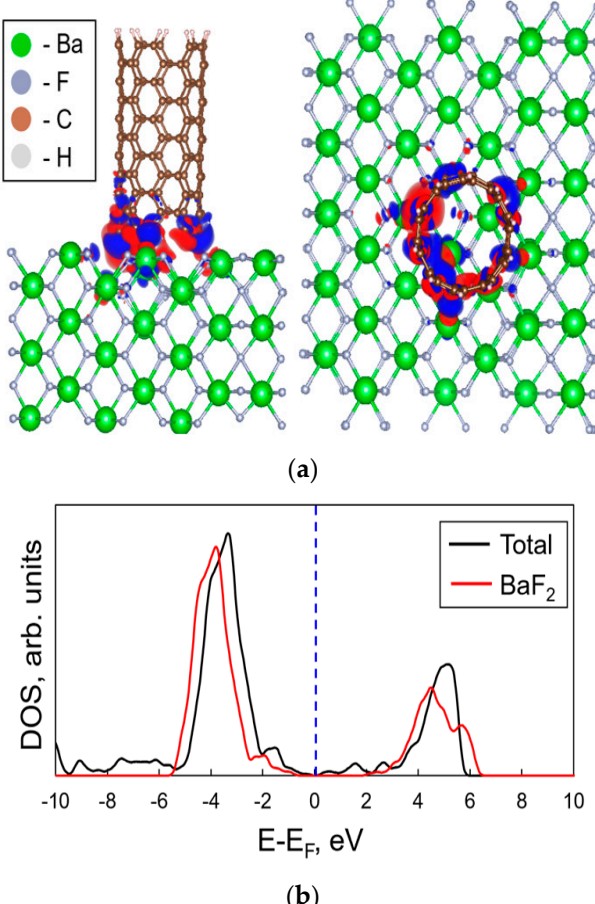

(**a**)

(**b**)

**Figure 6.** Atomic structure of the considered interface. (**a**) Distribution of the election density near the interface; (**b**) dependence of the electronic density of states on the energy for the CNT–BaF$_2$ interface (black line) and separately for the pure BaF$_2$ substrate (red line).

Relaxation of the interface structure shows that attaching of the CNT to the (100)-BaF$_2$ surface does not lead to significant changes of the interface structure. For this system the binding energy between the substrate and the CNT was calculated and found to be −5.36 meV/Å$^2$.

We also calculated the changes in the electronic properties caused by the presence of the deposited CNT. The distribution of electron density was plotted and the electronic density of states was calculated (see Figure 6b). Results indicate weak redistribution of electron density near the $BaF_2$/CNT interface (Figure 6, upper panel). Moreover, the dependence of the DOS on the energy also manifests about the weak interaction between the tube and the substrate (the shape of the main peaks remains, see bottom panel in Figure 6). Deposition of the CNT leads to the small shift of the electron density peak (around −4 eV) towards the blue region of solar spectra on ~1 eV. Deposition of the CNT also leads to the appearance of a certain number of energy levels close to the Fermi level, which can be seen from Figure 6.

All calculations were performed at the level of the density functional theory as implemented in the VASP package [24,25] within the augmented plane–wave basis set [26]. Due to the large size of considered system the plane–wave cutoff energy was set to 250 eV and the density of states was calculated only in the Γ-point. Several test calculations showed that decreasing the cutoff energy from 400 to 250 eV leads to relatively small changes in the total energy of the system.

Thus, slight interaction between carbon atoms and Ba and F atoms can be successfully considered and experimental data can be supported. Experimental and simulation results provoke and confirm the idea about the interaction at the structured materials interface under the laser-oriented deposition technique conditions.

## 4. Conclusions

Analyzing the obtained results, one can testify that the laser-oriented deposition technique has been applied in order to deposit the CNTs in possible vertical position at the inorganic crystal, such as the barium fluoride surface. The possible covalent bonding formation has been proposed to explain the established results. The proposed explanation is supported by the block scheme of the experimental set up, by the possible qualitative scheme to orient the CNTs via the electric field use, by the spectral experiments, AFM-image, wetting-angle measurements and via microhardness changes as well. It was obtained the increase of the transparency, wetting angle and hardness after CNT deposition on $BaF_2$ substrate. A quantum–chemical simulation was made to support the experimental data. The obtained results on the optimization of material properties based on $BaF_2$ can be useful for general optoelectronics purposes in the development of protective windows for the photosensitive diodes operating in the IR range of the spectrum; they are applicable to the space elements for protection from dynamic dust and are relevant for the elements of IR lasers. Besides different general application, which mentioned above, the $BaF_2$ substrates can be used in the liquid crystal modulator area as well (to align the liquid crystal dipoles) in order to switch or convert the IR radiation with good advantage due to the change of the wetting angle via nanostructuring.

It is important to note, that, indeed, it is necessary to continue the investigation of the inorganic crystal study by use the modulated scanning calorimetric approach, by X-ray analysis, paramagnetic resonance method and other approaches to confirm, for sure, the process of covalent binding of CNTS with the surface of inorganic materials, if a correlation is found between the diameter of nanotubes and the parameters of the materials elementary lattice. These experiments are planned for the future and will be shown in the subsequent publications on changing the properties of not only barium fluoride, but also calcium fluoride, potassium chloride and sodium via application of the LOD technique.

**Author Contributions:** Conceptualization, methodology, investigation, formal analysis and writing—original draft preparation were made by N.K. Spectral study and wetting-angle experiment was prepared by P.K. Quantum–chemical simulation was made by D.K. All authors have read and agreed to the published version of the manuscript.

**Funding:** This research received no external funding, but partially supported via the Vavilov State Optical Institute internal theme "LC-nanoWS$_2$" from 2018–2019. D.G.K. acknowledges financial support by Ministry of Science and Higher Education of the Russian Federation (project No. 01201253304).

**Acknowledgments:** The authors would like to thank their colleagues from the Lab for Photophysics of media with nanoobjects for the helpful discussion at the different steps of the research.

**Conflicts of Interest:** The authors declare no conflicts of interest.

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
