# Peer review of "Novel Perspective Coatings for the Optoelectronic Elements: Features of the Carbon Nanotubes to Modify the Surface Relief of BaF2 Materials"

_coatings, doi:10.3390/coatings10070661_

Round 1

Reviewer 1 Report

The article contains a rather important contribution to the development of novel perspective coating for the optoelectronic elements. Nevertheless, in its present form, it is absolutely not suitable for publication, since it does not contain a detailed analysis/overview of those new aspects that would have been necessary to be stated.

Introduction should be seriously revised and new references should be added to the reference list, which is rather short.  There are only 6 references (younger than 10 years old),  which clearly does not confirm the relevance of this work.

The article should attract new readers with its new ideas and only then will the article be cited in future and accordingly the impact factor of the journal will be increased.

Why carbon nanotubes? What new applications can be stimulated by this work ?  Why BaF2? 

Some possible applications and ideas:

BaF2 is known as one of the best fast scintillator (Kirm, M., Lushchik, A., Lushchik, C., Nepomnyashikh, A. I., & Savikhin, F. (2001). Dependence of the efficiency of various emissions on excitation density in BaF2 crystals. Radiation measurements33(5), 515-519).  Then, when carbon nanotubes are placed on BaF2 surface, this structure could be of great interest for high energy physics, as a nanostructured particle deflectors (CNT)  (Bellucci et al, 2006) with efficient BaF2 scintillator in the end.

Bellucci, S., Balasubramanian, C., Grilli, A. et al . (2006). Using a deformed crystal for bending a sub-GeV positron beam. Nuclear Instruments and Methods in Physics Research Section B: Beam Interactions with Materials and Atoms252(1), 3-6.

Finally, what about the long-term stability of these new coatings and their optical / radiation degradation?

Author Response

Dear referee!

Thanks a lot for your recommendations and questions. I have added the recently published papers in the references list and include the data about the BaF2 materials properties saving during 5 years. We have started to test these materials 5 years ago and controlled their properties in order to use them in the IR-laser scheme.

All changes are collared by yellow.

Thank you once again. Your recommendation really increases the paper.

Best Regards,

Natalia Kamanina

Reviewer 2 Report

This manuscript presents a laser-based deposition technique to orient carbon nanotubes in the vertical position to Here we used laser oriented deposition technique in order to modify the BaF2 surface. The manuscript also reports characterization results and theoretical calculations which are in a good agreement confirming the experimental results. The work presented in this manuscript is interesting and a significant contribution to the field and the community. This manuscript does convey the findings in a convincing manner, and the work presented in this manuscript is relevant to the Coatings journal. However, the manuscript has numerous grammatical and structural mistakes and the English language must be improved to be accepted for publication.

Author Response

Dear referee!

Thanks a lot for your recommendations and questions. I have seen mentioned by you the numerous grammatical and structural mistakes and the English language. All changes are collared by yellow.

Thank you once again. Your recommendation really increases the paper.

Best Regards,

Natalia Kamanina

Reviewer 3 Report

Authors used laser oriented deposition method to coat CNT on BaF2 substrate. Various property changes such as transmittance and contact angle were compared after coating of CNT.

Reviewer thinks that this paper is not organized well. It can distract readers's point of view. In addition, purpose of this work is not clear. Results are not presented in a scientific way. Authors claimed that strong change was observed in terms of transmittance and contact angle. However, the change of the transmittance is less than 2%, which is very small. Adding one more processing step for this change may not guarantee an advantage. Refractive index of CNT is close to 1.1 which is very close to that of air. There should be not much effect on optical property of BaF2 due to a very small difference of refractive index compared to air. Presentation of figure is also poor. 

Author Response

Dear referee!

Thanks a lot for your recommendations and questions. I have checked some paragraphs, which possibly are not explained well the results. All changes are collared by yellow.

Thank you once again. Your recommendation really increases the paper.

Best Regards,

Natalia Kamanina

Round 2

Reviewer 1 Report

After successful revision,  this paper can be accepted

Author Response

Thanks for your comments!